# Soft-Sensor System for Grasp Type Recognition in Underactuated Hand Prostheses

**DOI:** 10.3390/s23073364

**Published:** 2023-03-23

**Authors:** Laura De Arco, María José Pontes, Marcelo E. V. Segatto, Maxwell E. Monteiro, Carlos A. Cifuentes, Camilo A. R. Díaz

**Affiliations:** 1Telecommunications Laboratory (LabTel), Electrical Engineering Department, Federal University of Espírito Santo (UFES), Vitória 29075-910, Brazil; 2Federal Institute of Espírito Santo (IFES), Serra 29040-780, Brazil; 3Bristol Robotics Laboratory, University of the West of England, Bristol BS16 1QY, UK

**Keywords:** kinematic sensor, contact force sensor, grasp recognition, machine learning, optical fiber, hand prostheses

## Abstract

This paper presents the development of an intelligent soft-sensor system to add haptic perception to the underactuated hand prosthesis PrHand. Two sensors based on optical fiber were constructed, one for finger joint angles and the other for fingertips’ contact force. Three sensor fabrications were tested for the angle sensor by axially rotating the sensors in four positions. The configuration with the most similar response in the four rotations was chosen. The chosen sensors presented a polynomial response with $R^{2}$ higher than 92%. The tactile force sensors tracked the force made over the objects. Almost all sensors presented a polynomial response with $R^{2}$ higher than 94%. The system monitored the prosthesis activity by recognizing grasp types. Six machine learning algorithms were tested: linear regression, k-nearest neighbor, support vector machine, decision tree, k-means clustering, and hierarchical clustering. To validate the algorithms, a k-fold test was used with a k = 10, and the accuracy result for k-nearest neighbor was 98.5%, while that for decision tree was 93.3%, enabling the classification of the eight grip types.

## 1. Introduction

Around 57,802 people underwent an amputation in the first eight months of 2022 in Brazil [1], and in Colombia, approximately 533,051 people had some disability in their legs and arms in 2020 [2]. The main causes of upper-limb amputation are due to trauma, with a prevalence of 77%. The human hand is a powerful tool in the development of Activities of Daily Life (ADLs), and its loss causes a great psychological impact on the person [3,4]. The assistive devices, such as prostheses, which are a class of robotic hands, have as their main goal to lessen the psychological impact of the amputation and to support the performance of ADLs [5,6].

Having robotic hands as similar as possible to the human hand is a challenge, where it is intended to emulate functions such as grasping, holding, pushing, pulling, manipulating, and exploring [7]. The robotic hands could be classified as prostheses, industrial, social, and humanoid robotics depending on their potential application. In the case of the prostheses, the main objective is to replace the missing member [8] and they can be classified into three principal groups: aesthetic, body-powered, and electrically powered [9]. The aesthetic hands are passive devices whose main goal is to replace the lost limb physically [10]. The body-powered hands use harnesses and cables to capture the body movements and actuate the prosthesis. Finally, the electrically powered type uses electronic components such as motors to actuate the prosthesis. These prostheses are generally controlled by a Surface Electromyography signal (sEMG). However, transducers, servos, and potentiometers are other control signals [11]. In this last group, the principal development has been with rigid materials, with disadvantages such as high costs, large weights, and a lack of flexibility [12].

Recently, researchers have been working with a new line of prostheses based on soft robotics to overcome the disadvantages of commercial prostheses, such as thumb movement, high costs, low Degrees of Freedom (DoFs), and safety. These types of devices use materials with elasticity moduli more similar to the human body, improving their performance, adapting easily to the person using the device, and allowing the hand to have higher DoFs [12,13]. Among the differences between commercial and soft robotics prostheses, the most relevant is the number of DoFs, since it allows more movements that are similar to the human hand.

In addition, to improve the DoFs of prostheses, it is important to add sensors to the assistive device to improve control strategies and monitor the environmental variables [14,15]. Considering the recent advancements in the soft robotics field, the use of sensors based on optical fiber has increased due to its capabilities such as high flexibility, low costs, small size, low weight, robustness, biocompatibility, high sensitivity, and precision [16]. The optical fiber is a waveguide through which light propagates. Internally, the light is reflected along the fiber to be transmitted from one side to the other [17].

Some sensors commonly used in robotic hand instrumentation are contact force and bending sensors. For instance, Konstantinova et al. [18] developed a contact force and a proximity sensor in the fingertip. The sensors were tested in two fingers, and the assessment consisted of having one finger fixed and the other moving in the first finger’s direction. Between the fingers, an object was located at the fingertips’ height, and the second finger started to move in the object’s direction. The contact force sensor measured between 2 N and 12 N, and the proximity sensor detected the object at 20 mm. Yang et al. [19] used a fiber Bragg grating (FBG) sensor to identify when the robotic hand was grasping an object. In [20], a soft robotic manipulator was sensorized, and two types of sensors were developed based on polymeric optical fiber (POF): the first is a contact force sensor, and the other is a curvature sensor. The application is for a deep-sea robotic hand, where the sensors respond with outstanding performance underwater and in the air. Mori et al. [21] combined polyurethane with POF to produce a bending sensor for a pneumatic finger based on soft robotics. After fabrication, the sensor was located inside the flexible finger used in a robotic hand. The sensor information was used to control the finger’s performance. In [22], a curvature sensor was developed with polymeric optical fiber for monitoring the pneumatic fingers of a robotic hand.

Tactile sensors have been broadly used in slip detection. Feng et al. [23] proposed an FBG-based contact force sensor consisting of two arrays of six FBGs along the robotic fingers. Through different object movement simulations, the slip points were determined. The sensor information was used in a pattern recognition neuronal network to control the grasping of the robotic hand (accuracy 99.36%). In [24], the authors constructed a robotic hand with three fingers. Each finger was located in an array of eight FBGs to detect the contact position and force. Since it was possible to measure these variables correctly, the information could be used for controlling strategies. In [25], the authors instrumented a robotic finger in one manipulator. For this, an array of twelve FBGs was located between the palmar and dorsal sides of the finger. The palmar FBGs allowed the detection of contact with the object and the dorsal part to monitor postures. In [26], a proximity sensor with polymer optical fiber was used for haptic exploration with one robotic finger. In general, the sensor could detect objects. However, it had some limitations with small, narrow, dark, or highly reflective objects. In [27], a sensor to measure the applied force and an array of sensors to determine the force distribution was developed with polymer optical fiber. The sensor measured forces between 0 and 4.8 N and could detect the distribution correctly. One of the principal advantages of the sensor was that it could be miniaturized.

The human hand has the capability, through active exploration, to recognize objects for its manipulation without a visual aid. This ability is called haptic exploration. The recognition process could be through the somatosensory perception of patterns on the skin surface (sensing temperature, texture, slip, and vibration) or through the kinesthetic perception of limb movement, position, and forces [28]. Movements such as lateral motions, static contact, pressure, unsupported holding, enclosure, and contour following allow the brain to relate information to known object characteristics [29]. One of the main goals of the robotic hand is to make the device as close as possible to the member, so sensor and signal processing requires the capability of measuring and extracting information from the surrounding environment. For this, processing information with machine learning (ML) algorithms has played an important role in applications such as gesture recognition through sEMG signals, slip detection, and object recognition [30,31].

Therefore, object manipulation in robotic hands is one field concerning object recognition with machine learning approaches. Li et al. [32] instrumented two of four fingers with two bimodal sensors in the fingertips. One of the sensors measured contact pressure and environmental temperature, and the other measured thermal conductivity and object temperature. An artificial neural network was used to identify five objects with the sensors’ information, achieving accuracy of 95%. In [33], a tactile sensor was located on the robotic fingers’ phalanges (two sensors per finger). The sensors measured the pressure, gravity, angular rate, and magnetic field. The study recognized cylinders and square prisms in two situations. The first situation was the robotic hand performing a single grasp of the object, and the second was the manipulator performing exploratory movements of the object by displacing it with the thumb. Seven classifiers were tested in both situations: two random forests (RF), an extra-trees classifier, a ridge classifier, a support vector machine (SVM), a k-nearest neighbor (k-NN), and a multilayer perceptron (MLP). The random forest obtained the highest accuracy for the single and exploratory grasp with 50 decision trees, with 93.33% and 99.05%, respectively. The accuracy was improved with the exploratory movements for all algorithms.

The main goal of Konstantinova et al. [34] was a classification algorithm to distinguish between hard and soft objects. The manipulator consisted of two fingers; each finger was instrumented with four optical fiber sensors: one for force, one for torque, and the others for proximity. Three algorithms were tested: a voted perceptron, a ZeroR, and an SVM; the latter had the best result (accuracy of 87.3%). Kaboli et al. [35] instrumented the five fingertips of a robotic device with the tactile sensor BioTact to measure contact force. Three algorithms (SVM, passive-aggressive online learning, and expectation maximization) were tested with the texture of twenty objects. The SVM had the best result, with accuracy of 96%. In [36], the pneumatic fingers were instrumented with two sensors. The first was an optical fiber sensor to measure curvature, and the other was an intelligent digital display pressure transmitter to measure the input pressure of the finger. The study focused on recognizing gestures, object shapes, object sizes, and object weights, where four algorithms were tested (k-NN, SVM, logistic regression (LR), and k-means clustering (KMC)). The recognition pattern was tested with the information of the bending sensor and pressure sensor individually, and both sensors’ information. For the four pattern recognition tasks, the k-NN and KMC had higher accuracy; the first had the best results in most cases.

According to the state-of-the-art, optical fiber sensors assisted by machine learning algorithms have been used in tactile applications and object recognition. Although FBGs sensors are broadly used due to their multiplexing capabilities, one disadvantage remains in their interrogation systems, which are usually expensive and bulky, reducing their applicability [37]. In this context, cost-effective interrogation techniques [38,39,40] will allow the more effective integration of optical fiber sensors in soft robotics applications. As an alternative, polymer optical fiber (POF)-based sensors present the same advantages as silica fiber sensors but include higher mechanical robustness, more flexibility, easy manufacturing processes, and low-cost interrogation systems [41]. Measurements of variables such as angle [42] and pressure [43] with POF-based sensors have already been reported in the literature, suggesting easy integration with soft robotics applications.

The motivation of this work is to develop an intelligent soft-sensor system to add haptic perception to underactuated hand prostheses. This sensor methodology monitors the physical interaction during grasping activities to detect the object type grasped by a data-driven approach. The haptic sensor system developed here was implemented in the upper-limb soft robotics prosthesis PrHand (described in [44]). The soft-sensor system resembles the kinesthetic perception of the human hand by implementing two sensing modalities, finger joint angles and fingertip contact force measurements implemented in the prosthetic fingers. The sensor technology design is based on polymer optical fiber for both sensors. Three fabrications were tested by axially rotating the sensors in four positions for angle sensor development. The configuration with the most similar response in the four rotations was chosen. The contact force sensors were located at the fingertips to track the force made over the objects. For the machine learning implementation, 24 objects of the Anthropomorphic Hand Assessment Protocol (AHAP) related to eight grasp types were used. Six machine learning algorithms were tested; four were supervised (linear regression, k-nearest neighbor, support vector machine, and decision tree), and two were unsupervised (k-means clustering and hierarchical clustering). This article is divided into four main sections. Section 2 presents the angle and contact force sensor fabrication and setup characterization. Basic concepts of ML algorithms are also presented. Section 3 shows the results and the discussion of them. Finally, the conclusions and future works are depicted in Section 4.

## 2. Materials and Methods

### 2.1. Angle Sensor

For the angle sensor development, a polymer optical fiber (SH4001, Mitsubishi Chemical Co., Tokyo, Japan) was used due to its characteristics such as flexibility, impact resistance, high deformation, and low cost, which align with the prosthesis features. The working principle is the intensity variation, meaning that changes in the fiber curvature could be measured as voltage variations. Macro-bending on the POF leads to optical power leakage, which generates small variations on the photodetector (hundreds of micro-volts). Therefore, it is necessary to increase the sensor’s sensitivity [45]. A common approach is to polish a lateral portion of the fiber cladding and core to create a sensitive zone with a major quantity of optical power losses when bending this zone. The sensor’s sensitivity is highly influenced by the shape, length, and depth of this zone [45]. Thus, in the lateral section, there are increased power losses, and when there is bending in the opposite direction to the sensitive zone, which increases the power losses, it is possible to relate the power variation to the angle variation. Figure 1 shows the working principle for this sensor.

This configuration is broadly studied in the literature for angle measurement [17,46], where the sensitive zone is located on the bending side. However, this approach could be affected by misalignments. In this work, it is proposed to uniformly polish the fiber cladding and the core to mitigate the possible misalignment problems. First, the jacket (plastic coating) is completely removed from the bending zone, and a portion of the fiber cladding and the core is polished uniformly by rotating the POF on its own axis; see Figure 2. It is expected that rotations of the fiber do not affect the sensor response during the bending, i.e., the sensor’s sensitivity is constant at any axial rotation.

To verify the angle sensor response, three different sensor configurations were tested. The first was the side polish approach, where, to create the sensitive zone, a mini CNC 3018 machine (Generic) was used (Figure 3a). The fiber was located on the supports after the spindle motor turned the drill, and, finally, the bed closed the fiber to the drill until creating a 10 mm × 0.7 mm polish zone. For the second sensor, only the jacket was removed, with a 20 mm length to reduce mechanical stress. To remove the jacket, we used a 3D-printed structure that housed a razor blade over a hole where the fiber passed through it (Figure 3b). When the fiber was pushed, a cut along the jacket was created, where it was possible to extract the fiber. The coating was divided into equal parts, excluding the joint length, and it was located over the fiber again. The last sensor (called jacket remotion with cladding and core axial polish) followed the same process to remove the plastic coating. Moreover, the fiber was placed over the structure, as shown in Figure 3c, to polish the cladding and the core uniformly. The procedure consisted of a DC motor turning a pulley that, through a band, transmitted its movement to another pulley where the fiber was located. The polish was made with sandpaper when the fiber was turning around on its own axis.

To assess whether there are misalignment problems with the side polish approach and whether they are decreased with the new configuration, each fiber was located over the test bench depicted in Figure 4a. The system consisted of a light-emitting diode (LED) IF E97 (Industrial Fiber Optics, Tempe, AZ, USA), a phototransistor (PD) IF-D92 (Industrial Fiber Optics, USA), a servomotor MG995 (TowerPro, Shenzhen, China), and a 3D-printed structure to anchor the optical fiber. The LED and the PD were on opposite sides of the fiber, and the voltage in the PD was measured with a microcontroller Teensy 3.6 (PJRC, Honolulu, HI, USA). The servomotor was controlled by a microcontroller Arduino Uno (Arduino, Turin, Italy). Once the fiber was located in the system, the angle was changed every twenty seconds from 0° to 70° and back to 0° in steps of 10°. To analyze the sensor’s repeatability, the process was performed three times. The fiber was then rotated 90° and we repeated the abovementioned cycle. This process was repeated until the fiber gave an entire lap. For the first sensor, the side polish was positioned in parallel with the arm structure for the first position. Figure 4 shows the first position per sensor configuration: side polish, jacket remotion, and jacket remotion with cladding and core axial polish, from b to d, respectively. Once the sensor configuration was decided to be used, the prosthesis was instrumented, where the sensitive zone was located in the distal interphalangeal joint of each finger. To obtain more intensity variations, two sensitive zones per sensor were created.

The characterization of the first sensor (side polish) was described in previous works [47,48,49]. This sensor was already used to instrument the PrHand prosthesis, and it is used in this study for comparison purposes. The same protocol was used to characterize the new curvature sensor version (radial cladding and core polish without jacket protection). First, the sensors are anchored in the fingers. Following this, each finger performs cycles of opening and closing individually. The actuation mechanism of the prosthesis (Dynamixel servomotor) changed its angle to around 30° every 20 s from the finger entirely open to closed and back. Six cycles were performed, and the voltage in the PD was recorded with a Teensy 3.6 (PJRC, USA). As a reference system, a camera was used to record the finger movements. To track the angle changes, the open-source software Kinovea was used; for the tracking, it was necessary to locate three markers on the finger as reference points. The MATLAB software was used for the signal processing, which implemented a low-pass filter with a cutoff frequency of 0.5 Hz to smooth the signals. Spectral analysis was performed to define this threshold. The data were separated per motor angle and we obtained an average per point; this process was performed per finger.

### 2.2. Contact Force Sensor

For the contact force sensor, the same POF was used. However, the light coupling for this type of sensor is through side polishing instead of monitoring the optical power variations in the transmission mode. This means that, with this configuration, it is possible to measure the voltage in each end face of the fiber [50]. The voltage variations are generated by changing the distance between the LED and the sensitive zone. Therefore, as the LED is closer to the fiber, more optical power is coupled in the sensitive zone and vice versa. Thus, when a force is applied over the sensor, the LED is closer to the optical fiber’s sensitive zone, meaning higher light coupling, as is shown in Figure 5.

A 3D-printed mold was used for the sensor construction (Figure 6a). The chosen fiber polish was the lateral one since, in [49], it could measure the contact force in the desired range, between 0 N and 3 N. The CNC machine shown in Figure 3a was used for construction. To fabricate the sensor head, the LED WS2812 (WorkdSemi Co., Dongguan, China) and the fiber were located in the mold, with the side polish of the POF aligned with the LED’s lens; see Figure 6a. To stabilize the fiber over the LED and ensure that it did not undergo movement during the construction process, it was adhered to the cable of the LED with plasticine. After this, both elements were covered with transparent silicone, polydimethylsilicone (PDMS, WorkSemi Co., China). The demolding process was performed once the resin was dry. The contact force sensors with LEDs are shown in Figure 6b. The resin allows optical power to move from the LED to the fiber due to its transparency, and it also allows the fiber to stay or return to its original position when the force is applied and relaxed, respectively.

For the sensor characterization, the controlled compression structure of Figure 4 was 3D-printed and anchored on an optical breadboard. With this structure, it was possible to control the force applied over the sensor and how it was applied. A 263MXL Micrometer Head (Starrett, Athol, MA, USA) and a strain gauge LCM201 (Omega, Norwalk, CT, USA) were located in the structure. The micrometer allowed us to control and keep constant the applied force over the sensor. The strain gauge allowed determination of the applied force. A load cell amplifier HX711 (SparkFun, Niwot, CO, USA) and an Arduino Uno were used to measure the gauge information. For the characterization, the sensor was located in the structure, as shown in Figure 7b, and six cycles of compression and decompression were performed from 0 to 29.4 N, changing by 4.9 N every 20 s and allowing 10 s between the measurements until the voltage sensor stabilized. The range force value applied was defined considering a previous study where the maximum force that the PrHand prosthesis could achieve over an object was 30 N. The microcontroller Teensy 3.6 was used to measure the PD’s voltage variation.

The MATLAB software was used for the signal processing, which implemented a low-pass filter with a cutoff frequency of 0.5 Hz to extract the required information. To determine the cutoff frequency, a spectral analysis was performed where the main components of the signal were lower than 0.5 Hz. The data were separated per applied force and we obtained an average per point, and this process was performed for each finger.

### 2.3. Grasp Type Recognition Based on Machine Learning Algorithms

This section intends to offer a brief review of the ML algorithms used in this work to perform grasp type recognition. Four supervision algorithms were chosen to validate the proposed approach: linear regression, k-nearest neighbor, support vector machine, and decision tree. For the unsupervised algorithms, we chose k-means and hierarchical clustering.

#### 2.3.1. Algorithms

Linear Regression (LR): This is a supervised predictive model to obtain the relations between the variables. Equation (Equation 1) shows the relationship between the variables, where *a* and $b_{1},b_{2},\ldots ,b_{n}$ are constants, *y* is the independent variable, and $x,x_{2},\ldots ,x_{n}$ are the dependent variable, which is always a continuous value [51].
(1)$$y=a+b_{1}x+b_{2}x_{2}+\ldots +b_{n}x_{n}$$k-Nearest Neighbor (k-NN): The k-NN algorithm classifies the test sample based on its similarity with k samples in the training database. For this, the distance between the new sample and each training sample is calculated, and between the categories of the k training samples closest to the testing, the sample is found in the mode. To calculate the k value, the training data were divided into two equal parts and proven with k = 1, 2, 3, 5, …, m. To calculate the distances between the testing sample and the training sample, we used the Euclidean distance. See Equation (Equation 2), where $x_{1},x_{2},\ldots ,x_{n}$ was the training sample, $y_{1},y_{2},\ldots ,y_{n}$ was the testing sample, and *n* was the number of attributes.
(2)$$d=\sqrt{{\left(x_{1}-y_{1}\right)}^{2}+{\left(x_{2}-y_{2}\right)}^{2}+\ldots +{\left(x_{n}-y_{n}\right)}^{2}}$$Support Vector Machine (SVM): This algorithm is highly used because it creates a line or space to separate the database into its categories, allowing for the correct categorization of the new data. The line or space is called a hyperplane, and, for its creation, we use the extreme samples of each category as a limit.Decision Tree (DT): The decision tree is a supervised algorithm that creates rules to predict the value of the target variable. It has a hierarchical tree structure consisting of a root node, branches, internal nodes, and leaf nodes. DT learning employs a “divide and conquer” strategy by performing a greedy search to identify optimal split points within a tree. This splitting process is repeated top-down and recursively until all, or most, records have been classified under specific class labels.k-Means Clustering (KMC): This algorithm classifies the dataset into k clusters. Initially, the algorithm selects k random centroids after each sample is associated with the closest centroid, calculating the Euclidean distance between them. Once all the samples are classified into one centroid, a new centroid is calculated with the mean of all samples of each cluster. This process is repeated until the centroids have no changes.Hierarchical Clustering (HC): In the hierarchical clustering algorithm, the similar data of the dataset are clustering into a tree-like structure called a dendrogram. The algorithm works by adding the data samples in one cluster or dividing larger clusters into small ones. In the divisive hierarchical cluster, chosen for this work, all samples are in the same cluster, and the algorithm divides the group into subgroups until having one data point.

#### 2.3.2. Protocol

Considering that the main purpose of the prosthesis is to replace the missing member functionally, some tests allow the comparison of the prosthesis’ functionality concerning the human hand. One example is the Anthropomorphic Hand Assessment Protocol (AHAP) [52]. In particular, this protocol allows the comparison of different robotic hands by grasping different objects. This protocol was already implemented with the PrHand prosthesis in [44]. The assessment performed eight types of grasp with three objects per grip type. The main goal is to evaluate the similarity of each grasp concerning the theory about how the human hand performs each grasp type.

Due to the present work seeking to recognize the grasp type, the AHAP objects were evaluated since it is known how the prosthesis should grasp these objects. For this, we followed the steps presented in Table 1. Steps 1, 2, 4, and 5 had a duration time of 10 s, and step 3 had 20 s. The sampling frequency was 10 Hz, and steps 1, 3, and 5 were recorded with the Teensy 3.6, considering that steps 2 and 4 are for accommodation. The prosthesis grasped the object while looking up, since, in the first part of the AHAP protocol, the device receives the object in that position, the part where the grasping is evaluated.

Each step of the protocol was repeated three times per object. Following the AHAP study [52], the grip types that were evaluated were Hook (H), Spherical Grip (SG), Tripod Pinch (TP), Extension Grip (EG), Cylindrical Grip (CG), Diagonal Volar Grip (DVG), Lateral Pinch (LP), and Pulp Pinch (PP). The hook grasp objects were a skillet lid, a juice jar, and a bag. The objects for the spherical grip consisted of three styrofoam balls of three different diameters (75 mm, 96 mm, and 140 mm). A large marker, a tuna can, and a golf ball (a styrofoam ball of this size was used) were the tripos pinch objects. The extension grip objects were a plate, a medium box, and a small box. In the case of the cylindrical grip, the objects were a chip can, a coffee can, and a power drill. The diagonal volar objects were a bowl, a clamp, and a key. Finally, a small marker, a plastic pear (a styrofoam ball of this size was used), and a washer were part of the pulp pinch group. In Figure 8, we show an example of the prosthesis instrumented with the angle sensor and contact force sensor and grasping an object from the protocol.

#### 2.3.3. Algorithm Evaluation

The k-Fold Cross-Validation was used to assess the algorithm. The test consisted of dividing the database randomly into testing (30% data) and training (70% data) samples and calculating the accuracy. This process was performed ten times, considering that previous studies have low bias and modest variance with this number of repetitions [53]. To evaluate the algorithm’s performance, the accuracy was calculated.

To determine whether there was a significant difference between the algorithms’ results, a statistical test was implemented. First, we performed the Friedman test; the null hypothesis was that all the algorithms’ results were equivalent, with significance of 5%. If the *p*-value was lower than 0.05, the posthoc test was a Nemenyi test to compare all the algorithms between them.

## 3. Results and Discussion

### 3.1. Angle Sensor

The best response for the side polish configuration was to locate the fiber on the bent side (see dotted blue line in Figure 9a). Specifically, the 270° and 180° angles showed the importance of the alignment of the polish (sensitive zone) with the bent side, since the sensor’s behavior highly changes in these rotations (Figure 9a, dashed yellow line and squared gray line, respectively). In the case of 90° rotation, the sensor’s behavior was more as expected, but with lower sensitivity when compared to the 0° rotation. Therefore, the sensor performance is degraded if the fiber polish is misaligned with respect to the joint.

The second configuration, which was only the jacket remotion in the curvature zone, shows that the lack of jacket protection allows easy fiber deformation. Nevertheless, as there was no cladding and core axial polish, the sensitivity was low. Figure 9b shows the results. Moreover, it was found that from some angles, the voltage measured did not correspond to the expected behavior of the sensor. Specifically, for the rotations 0° and 180° (blue line with dots and gray line with squares, respectively, of Figure 9b). This behavior can be associated with environmental changes due to an alteration in the external light. Due to the fiber cladding being completely exposed and remaining transparent, the possibility to couple light from the environment is high.

The third sensor fabrication had the best results (see Figure 9c) due to the four axial rotation angles’ similar behavior. As compared with the other configurations, it can be seen that this approach mitigates the misalignment problems and improves angle detection. It is worth mentioning that the fabrication process is handcrafted, which does not guarantee uniform cladding and core polish around the fiber. This means that some cladding and core parts were more sanded than others, causing the sensor not to have exactly the same response for all angles. In addition, there is no guarantee that the fiber cladding and core will be perfectly concentric, contributing to this error. On the other hand, after the sanding process, the fiber cladding becomes diffuse and helps to filter the light’s effects and enhance the sensor’s sensitivity.

Since the third sensor configuration had the better response (Figure 9c), it was chosen to instrument the prosthesis fingers to measure the angle in the DIP joint. The results of the characterization graphs per finger are shown in Figure 10. Moreover, we depict the error bars corresponding to the standard deviation for each angle value. As the characterization was performed over the prosthetic fingers, it was difficult to ensure that each opening/closing cycle was performed in the same way, generating errors in the angles and in the voltage, which is expected due to the nature of soft robotic prostheses construction [44]. It is observed that with increments, the error in the angles proportionally increases the error in the measured voltages. In addition, it is worth mentioning that the measurement of the angles was performed with the Kinovea software, which followed the prosthetic finger’s path during the test. However, it was necessary to correct manually some angles as the measurement was not correctly detected.

The equations for the closing and opening process per finger are presented in Table 2, where $V_{c}$ is the closing voltage, $A_{c}$ is the closing angle, $V_{o}$ is the opening voltage, and $A_{o}$ is the opening angle. All sensors presented a polynomial behavior, and the $R^{2}$ per equation is also presented. The lowest $R^{2}$ was for the index finger opening equation with 92%, but most were higher than 98%, implying that all sensors had a good relationship between the angle and voltage changes. Due to the DOFs and complex movements of the fingers, when compared with rigid structures, nonlinear behavior is expected. The polynomial sensor response associated with the highest angles (when the fingers are opened) presents higher voltage variations due to the fiber having more movement freedom (closing cycles). In the opposite case, when the fingers are closed, the fiber movement is restricted, and less deformation is applied to the fiber, leading to lower voltage changes (opening cycles). In addition, due to the Young’s modulus of the fiber being higher than the prostheses’ tendons, there is a difference in the behavior of the fingers in each cycle, both closing and opening. Moreover, mechanical issues can contribute to the fingers not returning to their initial positions. The worst cases are for the index and thumb fingers. As the grasp type recognition is performed with machine learning algorithms, the fact of having polynomial sensor responses is not a problem.

Table 2 presents the hysteresis percentage per finger. The hysteresis is attributed to the flexible character of both the fiber and the prosthetic fingers. Therefore, it was difficult to ensure that the opening and closing cycles followed the same path; the worst case was 17% for the little finger. Leal et al. have proposed techniques to compensate for the hysteresis and the viscoelastic effects [42], which can improve the angle estimation. The sensors showed a voltage variation of between 250 and 600 mV with respect to the angle. Considering this, using analog-to-digital conversion (ADC) with a high resolution is useful. The microcontroller Teensy 3.6 has an ADC with a 16-bit resolution. Since the sanding process was handmade, this meant that not all fibers had the same intensity ranges; in addition, each finger had different movement, leading to different responses.

**Table 2 sensors-23-03364-t002:** Angle voltage equations per finger.

Finger	Closing	R^2^ (%)	Opening	R^2^ (%)	H (%)
Little	V${}_{c}$ = 0.0001A${}_{c}$${}^{2}\;-\;$0.0189A${}_{c}$ + 1.5424	98.81	V${}_{o}=9\times 10\;-\;$5A${}_{o}$${}^{2}$ + 0.0154A${}_{o}$ + 1.5289	99.96	16.62
Ring	V${}_{c}$ = 0.0002A${}_{c}$${}^{2}\;-\;$0.0225A${}_{c}$ + 1.6044	93.98	V${}_{o}$ = 0.0003A${}_{o}$${}^{2}\;-\;$0.0247A${}_{o}$ + 1.6698	99.33	13.60
Middle	V${}_{c}\;=\;-2\times 10\;-\;$5A${}_{c}$${}^{2}\;-\;$0.0005A${}_{c}$ + 2.6976	99.79	V${}_{o}=\;-3\times 10\;-\;5$A${}_{o}$${}^{2}$ + 0.0007A${}_{o}$ + 2.6801	99.78	2.81
Index	V${}_{c}$ = 0.0002A${}_{c}$${}^{2}\;-\;$0.0347A${}_{c}$ + 2.4522	99.24	V${}_{o}\;=\;-9\times 10\;-\;$5A${}_{o}$${}^{2}\;-\;$0.0048A${}_{o}$ + 1.6853	92.03	13.18
Thumb	V${}_{c}\;=\;-5\times 10\;-\;$5A${}_{c}$${}^{2}$ + 0.0064A${}_{c}$ + 2.9612	99.84	V${}_{o}\;=\;-0.0002$A${}_{o}$${}^{2}$ + 0.0356A${}_{o}$ + 1.3477	98.61	9.19

### 3.2. Contact Force Sensor

In Figure 11 are shown the characterization results per contact force sensor. The blue dotted lines represent the compression behavior, and the dashed orange ones reflect the decompression. In addition, each graph depicts the error bars, which represent the standard deviation per point. This sensor had less error with respect to the angle sensor since the micrometer structure (see Figure 7) allowed the application of a controlled force. However, since the applied force was changed manually, some errors could occur during the process of characterization.

In general, the sensor was easy to fabricate, but the position of the LED with respect to the fiber side polish highly influences the behavior of the sensor response. Among the construction variables that could affect the sensor behavior, two principal groups were identified. First, although the side polish was performed with a CNC machine, not all sensors were exactly the same because of the eccentricities of the core/cladding and the jacket, which did not guarantee the same material polished, leading to changes in the sensor’s response. The second group was related to the sensitive zone and the LED position. On the one hand, the distance between the two elements caused the sensor’s working range to be larger. In addition, the LED position with respect to the side polish allows more or less optical power, depending on the LED angle with respect to the sensitive zone.

The equations for the compression and the decompression per finger are presented in Table 3, where $V_{c}$ is the compression voltage, $F_{c}$ is the compression force, $V_{d}$ is the decompression voltage, and $F_{d}$ is the decompression angle. In Table 3 is also presented the $R^{2}$ per equation; three sensors presented polynomial behavior, and the other two were linear. The lowest $R^{2}$ was for the index finger opening equation with 95%. Nevertheless, the majority were higher than 98%, implying that all sensors had a significant relationship between the applying force and voltage changes. Moreover, Table 3 shows the hysteresis percentage per finger caused by the elastic deformation of the silicone and the fiber. The two worst cases are 24% for the little finger and 21% for the thumb; for the others, the result was less than 8%.

**Figure 11 sensors-23-03364-f011:**
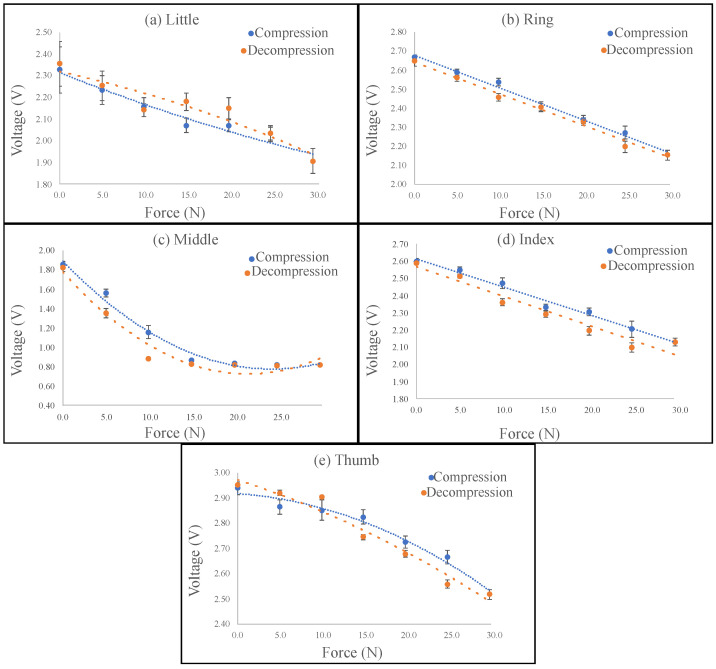
Contact force sensor characterization per finger: the dotted blue line represents compression, and the dashed orange line reflects decompression. (**a**) Little. (**b**) Ring. (**c**) Middle. (**d**) Index. (**e**) Thumb.

There are some parameters that are related to the sensor’s sensitivity in terms of measure, such as the signal-to-noise ratio (SNR) and the ADC resolution. If there is a poor SNR or lower ADC resolution, the signal variations as a function of either curvature or pressure can be hidden by the noise or by the same quantification value, presenting no variation in the voltage measured by the ADC. Therefore, it is expected that the lower the sensitivity, the higher the possibility of not detecting anything. For this reason, we selected an ADC with a 16-bit resolution. We performed digital signal filtering before applying the machine learning algorithms. Despite most angle and contact force sensor responses being polynomial, the sensors’ sensitivity was sufficient to have different sensor responses per object, as long as there were objects with different sizes.

### 3.3. Grasp Type Recognition Based on Machine Learning Algorithms

Despite 200 samples being acquired in step four, presented in Table 1, only 100 samples were used per repetition and object due to the signals presenting noise. Thus, the database consists of 7200 samples in total. The first algorithm tested was linear regression. Figure 12 shows the confusion matrix per type of grasp. The best results were the cylindrical grip, pulp pinch, and extension grip (33.6%, 32.7%, and 30.7%, respectively). However, the accuracies were lower than 50%. The probability of incorrect classification with this algorithm was very high, so it was concluded that these data could not be considered for the classification with the data of the angle sensor and contact force sensor presented here. The k-fold result (k of 10) was 21% with a standard deviation of 0.2.

The second algorithm used for the test was the k-NN. First, we chose the k with which the algorithm had the highest accuracy. For this, the training data were divided into two smaller datasets; the first division corresponding to 70% (3584 samples of the 5040 training samples) was used for training and the other 30% for testing with different k neighborhoods. In total, we tested 20 k from 1 to 20 in steps of 1, but, from k higher than 4, the accuracy was the same. Table 4 shows the accuracy results up to k equal to 7. The best results were for k equal to 1, 2, and 4. The k equal to 4 was chosen due to considering more neighbors than the other k with the same accuracy.

The confusion matrix with the k-NN results for grasp type recognition is shown in Figure 13. All the accuracies per grasp were higher than 96%. The k-fold result was 98.5% with a standard deviation of 0.01. Moreover, the prosthesis does not perform all the grasps as the theory indicates, and it was possible for this algorithm to identify the different grip types.

The following algorithm tested was the SVM, whose confusion matrix is shown in Figure 14. In this case, only the hook grasp was well classified, and the other ones were also classified as hook grasp. In the algorithm’s implementation, it was possible to see that the data were always classified as the object H1, so this algorithm could not classify the grasp types correctly. The k-fold result was 12.5% with a standard deviation of 0.0.

Regarding the decision tree algorithm (See Figure 15), the average accuracy result was 94.3% with a standard deviation of 4.8. This algorithm had good results since all the accuracies per type of grasp were higher than 88%. This had lower accuracy than the k-NN and a higher standard deviation. The greatest differences are in the results of the hook and the pulp pinch. However, it is an option for the recognition of grasp types.

The last two tested algorithms were unsupervised, and both had poor results since the k-fold result was 10.3% ± 5.6 for the KMC and 12.5% ± for the hierarchical clustering. Figure 16 and Figure 17 show the accuracy per type of grasp. The best classification was the lateral pinch with 21%, so it was concluded that this algorithm is not an option for grasp type recognition. The same result was achieved with the HC as compared to the SVM algorithm, i.e., this always classified all grasps as hook grasp; therefore, it was not possible to use this algorithm to classify the different grasp types.

From the LR results, we inferred that the data did not have linear behavior since the accuracy was 21%. Moreover, the support vector machine results determined that the data were mixed and the parameters were not separated enough to form a plane to divide them. With the unsupervised algorithms’ results, it was inferred that there was no clustering formation of the data in both; therefore, hierarchical clustering and k-means clustering need data organization to perform better. With the accuracy of the decision tree algorithm, it was concluded that it had the best performance relative to those described above, as it considers the training data to create rules to classify the data.

Table 5 shows the k-fold result per algorithm. A Friedman test was implemented to ensure that a statistical difference existed between the algorithms’ results. The *p*-value was 1.04 × 10${}^{-8}$; as it was less than 0.05, it was concluded that at least one of the algorithms had significant differences from another. A Nemenyi test was performed as the posthoc test, calculating the CD value of 2.38. As all the ranks had a difference higher than 4, it was concluded that there were significant differences between the algorithm’s results. The k-NN had the best performance, followed by the decision tree.

Considering the obtained results of the algorithms, a Principal Components Analysis (PCA) was performed to visualize the database; the result is shown in Figure 18. It is possible to see that most of the data were mainly focused on one area, and the clustering formation was low. This could affect the accuracy of unsupervised algorithms since clustering formation is essential for correct classification. Moreover, it was observed that the data did not have linear behavior, which caused the linear regression algorithm to have low accuracy.

Analyzing the confusion matrix of the k-NN algorithm, there were a few occasions when the hook grasp was classified as a diagonal volar grip and spherical grip as an extension grip. These errors were associated with some limitations in the DoFs of the prosthesis, i.e., the fingers did not bend as expected, leading to poor classification. In addition, the spherical grip was sometimes classified as a cylindrical grip, which could be associated with the geometric shape of both groups being related to circles. Therefore, some samples might be misclassified. In the tripod pinch cases, the objects with circular shapes could be classified wrongly as spherical. The lateral pinch classified as diagonal volar could be due to both grips being similar due to the fingers’ location. The principal error with diagonal volar grip was identified in the pulp pinch. The PP was a complex type of grasp for the prosthesis, since the grasp needed the union of the thumb finger with the index finger, which was not possible for the prosthesis, which could result in the prosthesis grasping the object similarly to the DVG.

In a preliminary study, the angle sensors with side polish were used to instrument the PrHand prosthesis, and their information was used to recognize four of the eight grasp types (H, SG, TP, and CG). The k-NN algorithm was used in the study, and the k-fold accuracy result was 92.81 ± 0.47% [48]. Comparing the results of this study with the last one, this study had better results since the accuracy was 98.5 ± 0.01%. Thus, improving the angle sensor and including the contact force sensor improved the results. It was noticed that the hook grasp accuracy was lower for this study; the error was associated with the number of grasp types tested, considering that were eight grasp types and the additional grasp types had greater similitude with the hook grasp.

Table 6 shows the results obtained in this study concerning the literature. In the state-of-the-art, one of the most common algorithms implemented and with great results was the SVM, with accuracy of 96% according to Kaboli et al. [35] and 87.3% according to Konstantinova et al. [34]. For these studies, the used sensors are not based on optical fiber. In [35], the sensor used measured the contact force with 19 electrodes, meaning more inputs, which could lead to the better response of the algorithm in comparison with the one implemented in this study. In the case of [34], it detected only two variables, namely whether the object was hard or soft, meaning that there were only two variables to identify. The spaces that separated the information into categories might have been better formatted for these two variables. In [36], two sensors were used in recognizing the gesture, object shape, size, and weight. For this study, an optical fiber sensor curvature was used. The results showed that the k-NN classifier and the KMC algorithm had the best results, the first with the best accuracy in most cases. Comparing the shape recognition result (the most similar to grasp types) concerning this work, the last one had the best k-NN algorithm accuracy. Thus, it is implied that the contact force sensor allowed better algorithm performance for the presented study.

The research community has been working on the use of sensor systems to collect human data and classify the information according to grasp types. Souza et al. [54] worked with a tactile sensor to take data from a human hand with a glove and identify grasp intention. Moreover, as a continuation of their work, El-Khoury et al. [55] created a model with sensor information for controlling a robotic hand. Using sensors such as the one developed here could offer a cheaper alternative to this recognition system, with the mentioned fiber optic’s advantages. Both sensors have great potential to be included in soft robotics devices and in devices to capture human movement.

## 4. Conclusions

Considering the necessity to obtain devices with similar capabilities to the lost limb, the development of technology that emulates the body’s sensors is needed. In this work, a system that allowed the PrHand prosthesis to obtain haptic perception capability was developed and tested. For this, considering kinesthetic perception, two sensors were developed. The first, the angle sensor, was compared regarding the ability to sense member movement, since it allows measurement of the fingers’ angles. The second, the contact force sensor, was compared regarding the ability to sense applied forces, since it measures the forces that the PrHand applies over the object. Since the prosthesis is based on soft robotics, fiber optics, due to its flexibility, is a relevant candidate in measuring the parameters that control prosthesis functioning. The advantages regarding resistance to electromagnetic interference, flexibility, and low cost are very important in this type of development, especially because one of the main objectives of this prosthesis is a low cost.

Moreover, side polishing has been broadly explored in the literature; this method of fabricating sensors could cause misalignment problems that affect the sensor behavior, as shown in the experiment where the fiber was rotated. In the configuration of the jacket remotion fiber, the lack of jacket protection allows for easy fiber deformation. However, the sensor’s sensitivity was not as high when cladding and core polish were performed. The fabrication of the sensor that improved the misalignment problems was jacket remotion with cladding and core axial polish, whereas the sensor for different rotational axial angles had a similar response.

Regarding the sensor performance, it was concluded that all angle sensors’ behavior was well described with polynomial equations that described the relationship between the angle and the voltage variations, since the $R^{2}$ were higher than 92%. Due to the fiber and prosthetic fingers’ flexibility, it identifies a response for closing and another for opening, with a maximum hysteresis of 17%. This could be compensated with some techniques described in the literature [45] and by improving the sensor fabrication and the instrumentation of the prostheses. An easy-to-fabricate and low-cost sensor is generally suitable for the PrHand prosthesis’ characteristics.

The contact force sensors’ responses were well described with the equation, since all $R^{2}$ were higher than 95%. Two sensors were described with linear responses, and the other three with polynomials. Due to the flexible characteristics of the sensor, there were two sensor responses (one for compression and the other for decompression), leading to hysteresis in most cases less than 10%. However, two sensors, thumb and little, had hysteresis of 21% and 24%, respectively. This effect could be compensated with some techniques from the literature. The fabrication process of the sensor was easy; however, it was important to control as much as possible all the external variables that could affect the construction process, since, in the curing process, the fiber could undergo movements that affect the sensor’s behavior. This was the reason that some of them had a polynomial relationship with the voltage and others had a linear one.

After the sensors’ fabrication and characterization, the PrHand prosthesis was instrumented to obtain information about the environment (angle and force per finger). With sensor information, we tested six algorithms, among which the linear regression, support vector machine, k-means clustering, and hierarchical clustering did not yield good results, since their accuracies were lower than 22%. This was associated with the lack of clustering formation in the data, verified with the PCA results. The other two algorithms, k-NN and decision tree, had accuracies of 98.5% and 93.3%, respectively. The classification label was given for these algorithms by considering the training data labels.

In the literature review, the SVM was one of the most used ML algorithms, but it yielded was one of the worst results for this study. It was concluded that the information of the sensor highly influenced the correct behavior of the algorithm, since the sensor gives contact force information. Nevertheless, it is a more robust sensor than the one implemented in this study considering the 19-electrode array that provides the information. Moreover, one study that recognized different objects’ characteristics had great results with k-NN, agreeing with the results of this study.

One of the limitations of this study is the contact force sensor’s construction, since, during this process, it was challenging to control all the variables during the curing process. Moreover, it is necessary to evaluate whether the PMDs are the best option for this sensor because, after a while, the silicone starts to break, so the lifetime of the sensor is not as high as expected. In addition, for the characterization of the angle sensor, as it was performed on the fingers, it was difficult to ensure that, for all trials, the fingers always closed in the same way, causing some errors.

## Figures and Tables

**Figure 1 sensors-23-03364-f001:**
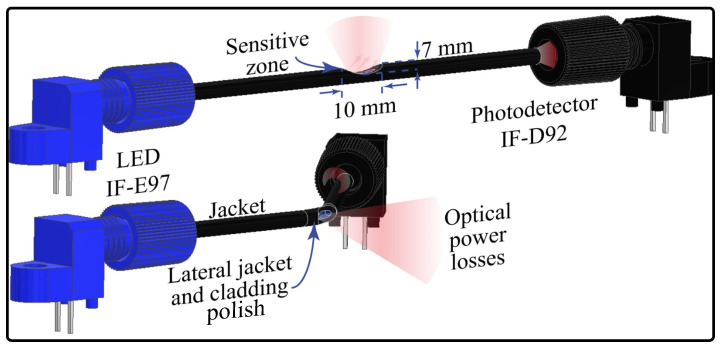
Working principle: side-polished angle sensor.

**Figure 2 sensors-23-03364-f002:**
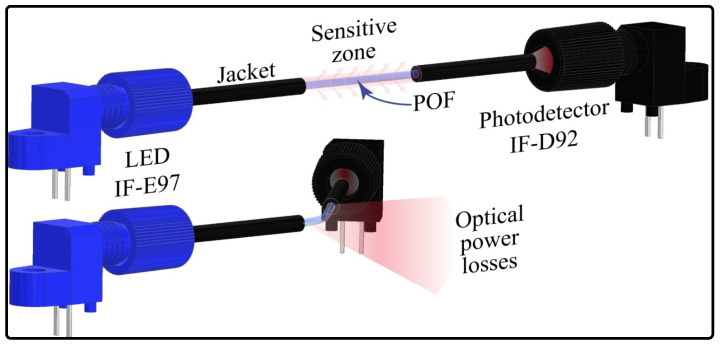
Working principle: jacket remotion with cladding and core axial polish angle sensor.

**Figure 3 sensors-23-03364-f003:**
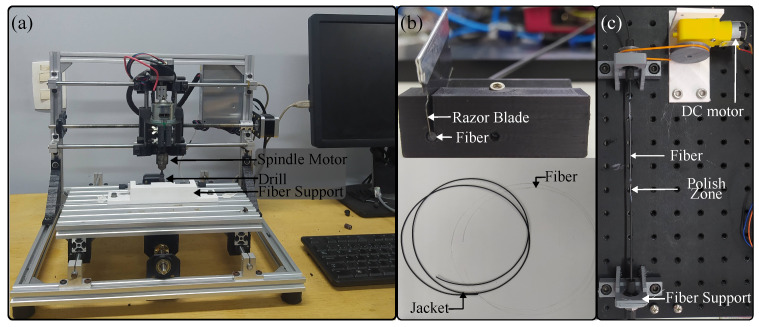
Devices used in the sensor’s construction. (**a**) CNC. (**b**) 3D-printed structure and fiber cladding separated from its jacket. (**c**) 3D-printed structure to obtain the polish around the fiber.

**Figure 4 sensors-23-03364-f004:**
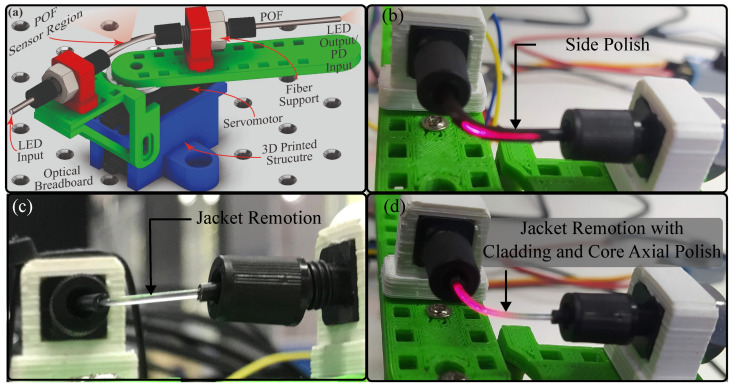
Test bench for the angle sensor and first position per fiber configuration. (**a**) Test bench CAD. (**b**) Fiber sensor with side polish. (**c**) Fiber sensor jacket remotion. (**d**) Fiber sensor jacket remotion with cladding and core axial polish.

**Figure 5 sensors-23-03364-f005:**
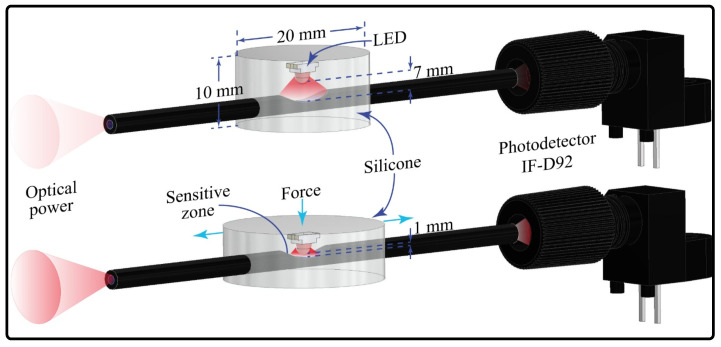
Working principle of contact force sensor.

**Figure 6 sensors-23-03364-f006:**
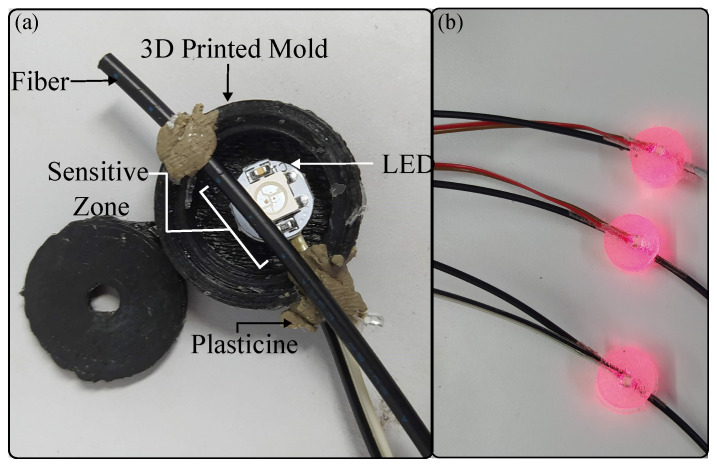
Construction of the contact force sensor. (**a**) 3D-printed mold. (**b**) Contact force sensor.

**Figure 7 sensors-23-03364-f007:**
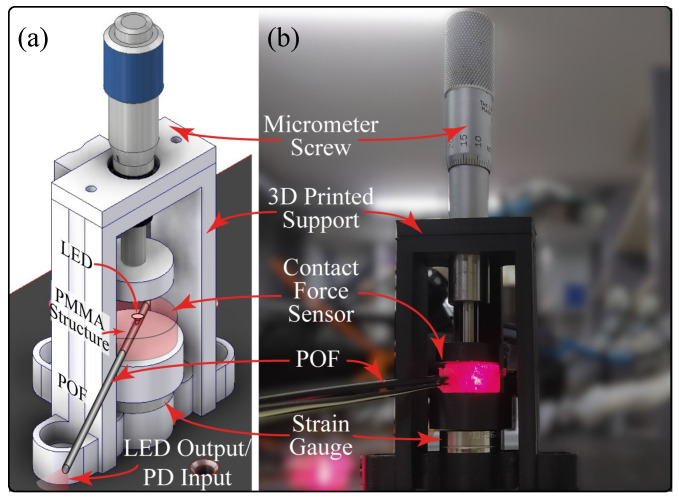
Test bench for the contact force sensor. (**a**) CAD. (**b**) Real setup.

**Figure 8 sensors-23-03364-f008:**
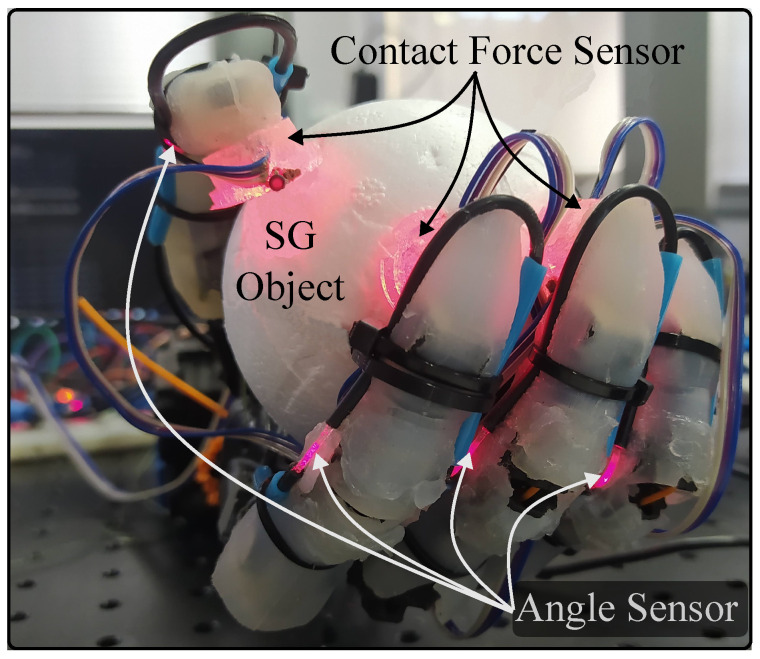
Prosthesis with the soft-sensor integrated, holding one object for the protocol.

**Figure 9 sensors-23-03364-f009:**
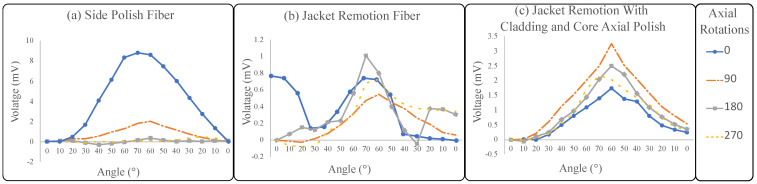
Results of the three sensor configurations per rotation: the dotted blue line corresponds to 0°, the dotted and dashed orange line corresponds to 90°, the squared gray line corresponds to 180°, and the dashed yellow line corresponds to 270°. (**a**) Side polish fiber. (**b**) Jacket remotion. (**c**) Jacket remotion with cladding and core axial polish.

**Figure 10 sensors-23-03364-f010:**
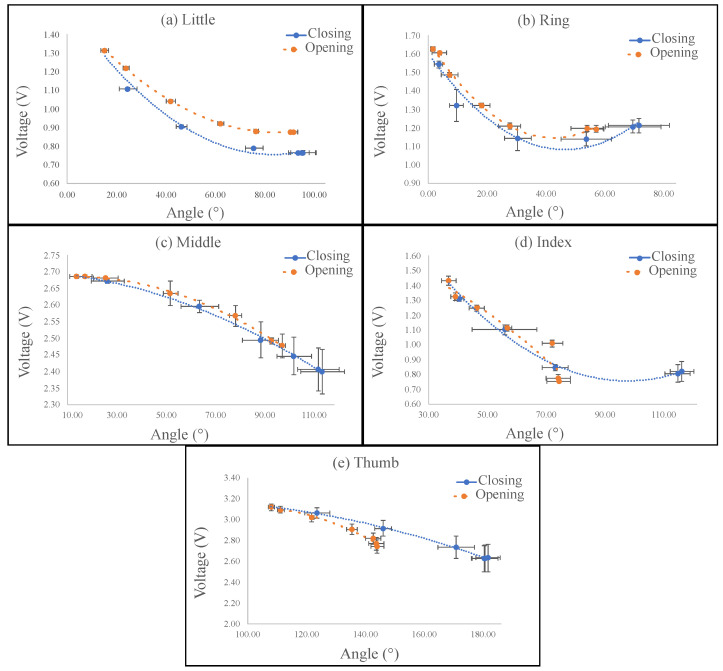
Angle sensor characterization per finger: the dotted blue line represents the closing and the dashed orange line the opening. (**a**) Little. (**b**) Ring. (**c**) Middle. (**d**) Index. (**e**) Thumb.

**Figure 12 sensors-23-03364-f012:**
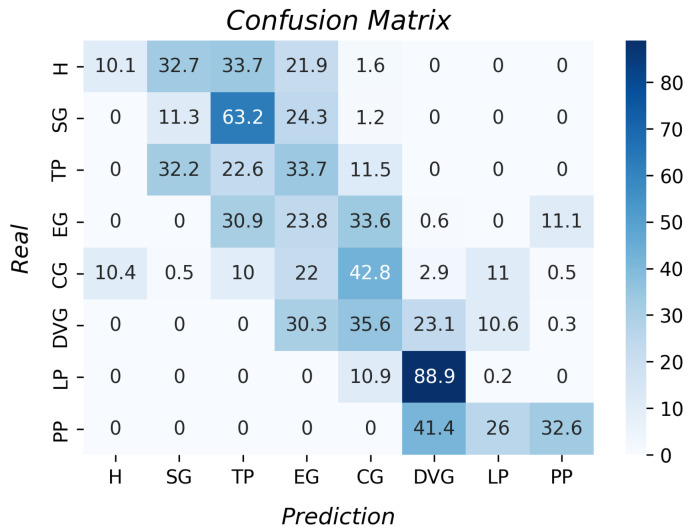
Accuracy per type of grasp for the linear regression algorithm.

**Figure 13 sensors-23-03364-f013:**
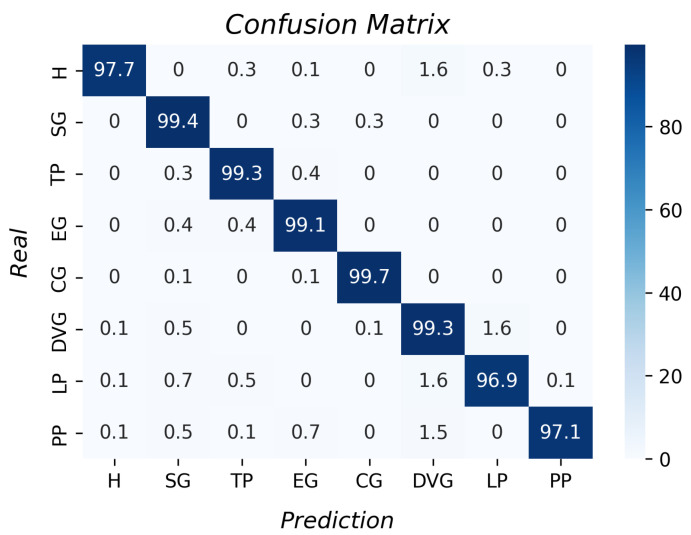
Accuracy per type of grasp for the k-NN algorithm.

**Figure 14 sensors-23-03364-f014:**
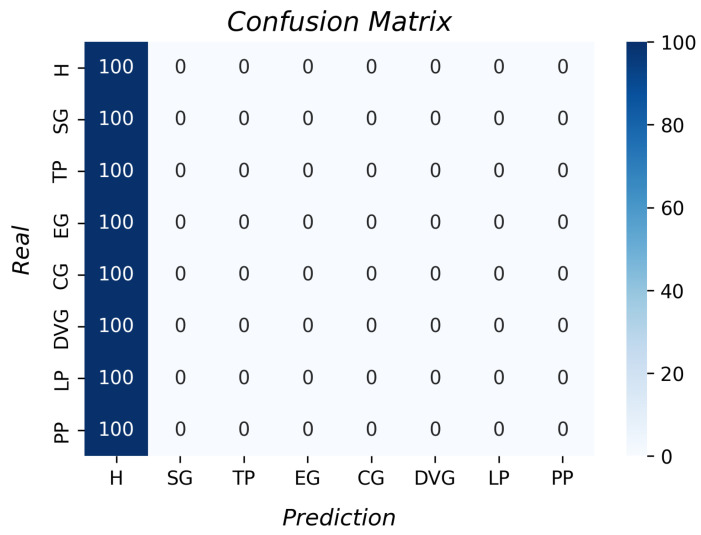
Accuracy per grasp type for the support vector machine algorithm.

**Figure 15 sensors-23-03364-f015:**
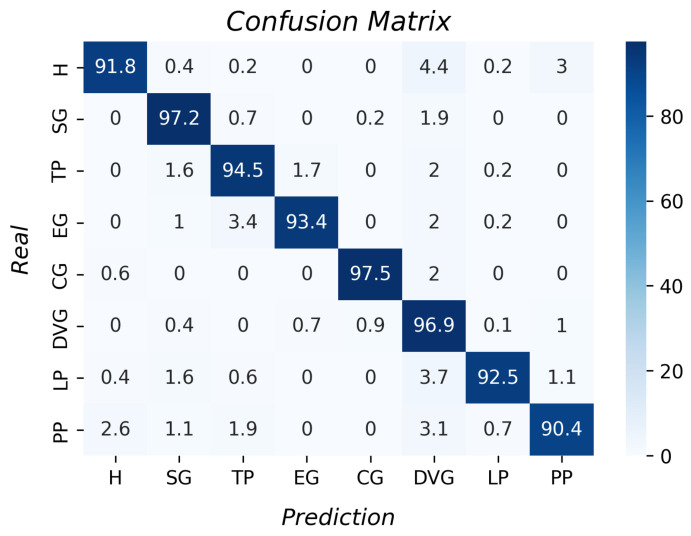
Accuracy per type of grasp for the decision tree algorithm.

**Figure 16 sensors-23-03364-f016:**
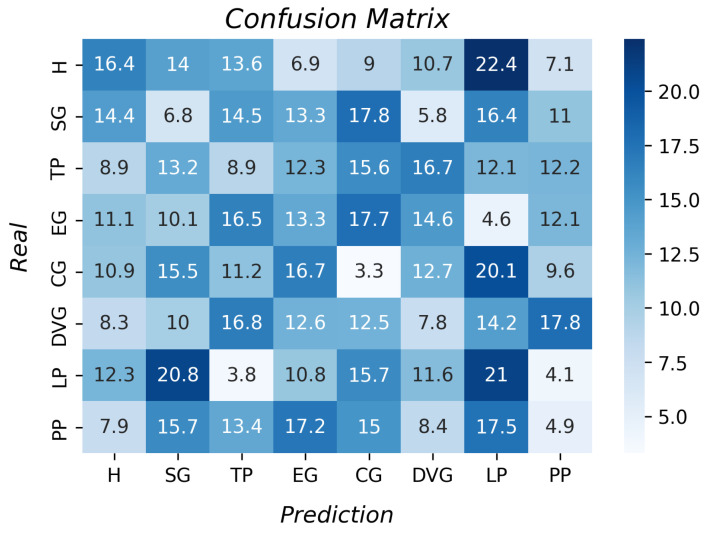
Accuracy per type of grasp for the KMC algorithm.

**Figure 17 sensors-23-03364-f017:**
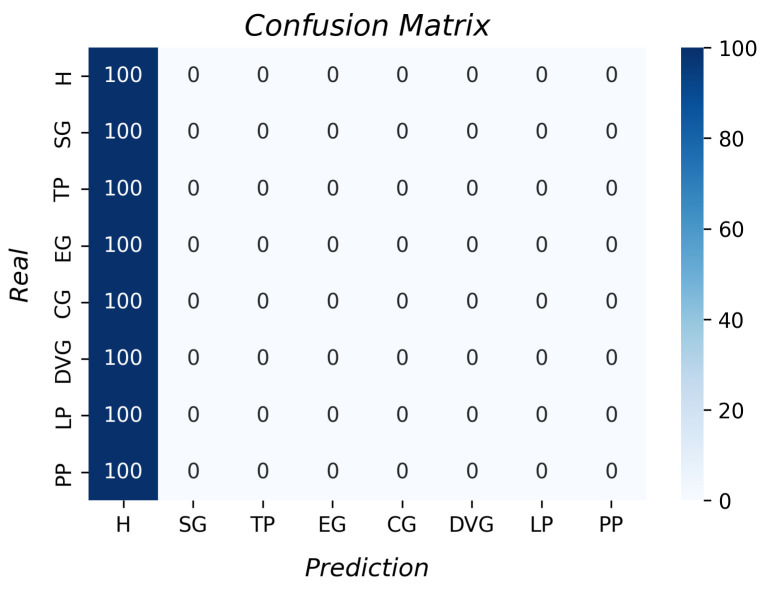
Accuracy per type of grasp for the hierarchical clustering algorithm.

**Figure 18 sensors-23-03364-f018:**
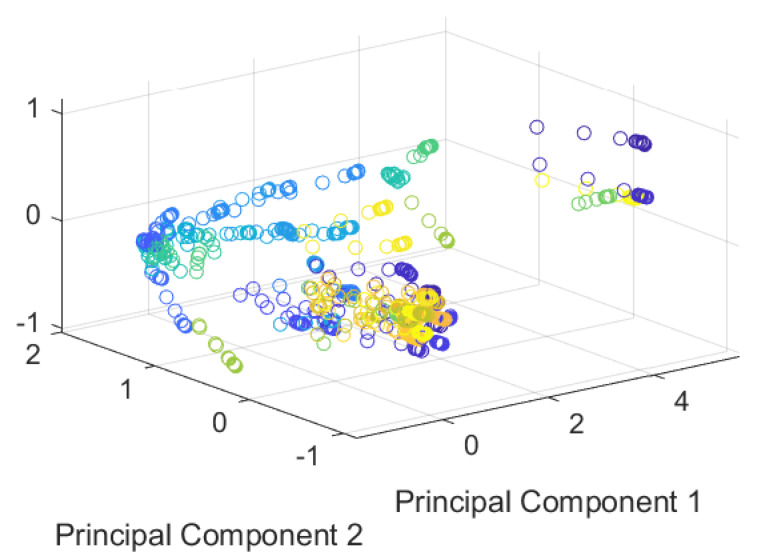
Principal Components Analysis (PCA).

**Table 1 sensors-23-03364-t001:** Steps for the acquisition protocol.

Step	Description
1	The prosthetic hand is open
2	Place the object and close the prosthesis
3	The prosthesis holds the object
4	Open the prosthesis and remove the object
5	The prosthetic hand is completely open again

**Table 3 sensors-23-03364-t003:** Contact force voltage equations per finger.

Finger	Compression	R${}^{2}$ (%)	Decompression	R${}^{2}$ (%)	H (%)
Little	V${}_{c}$ = 0.0001F${}_{c}$${}^{2}\;-\;$0.0158F${}_{c}$ + 2.3138	95.53	V${}_{d}=-$0.0001F${}_{d}$${}^{2}$ + 0.00864F${}_{d}$ + 2.3178	99.96	24.45
Ring	V${}_{c}$ = 0.0173F${}_{c}$ + 2.6786	99.16	V${}_{d}=-$0.017F${}_{d}$ + 2.6428	99.01	0.03
Middle	V${}_{c}$ = 0.0019F${}_{c}$${}^{2}-$0.0928F${}_{c}$ + 1.8887	98.47	V${}_{d}=-$0.0023F${}_{d}$${}^{2}$ + 0.0993F${}_{d}$ + 1.7779	95.8	3.40
Index	V${}_{c}=-$0.0165F${}_{c}$ + 2.6126	98.59	V${}_{d}$ = −0.0173F${}_{d}$+2.5671	94.75	8.04
Thumb	V${}_{c}=-$0.0004F${}_{c}$${}^{2}-$0.0022F${}_{c}$ + 2.9164	97.74	V${}_{d}=-$0.0002F${}_{d}$${}^{2}-$0.0109F${}_{d}$ + 2.972	96.69	21.06

**Table 4 sensors-23-03364-t004:** k tested for selection for the k-NN algorithm.

k	1	2	3	4	5	6	7
Accuracy	97.47732	97.47732	97.44898	97.47732	97.44898	97.44898	97.44898

**Table 5 sensors-23-03364-t005:** k-fold result per algorithm.

	Accuracy	Standard Deviation
LR	21.1	0.2
k-NN	98.5	0.01
SVM	12.5	0.0
DT	93.3	0.2
KMC	10.7	5.9
HC	12.5	0.0

**Table 6 sensors-23-03364-t006:** Comparison of the results obtained in this study concerning the literature.

Name	Objective	Number of Sensors	Type of Sensors	Sensing Variables	Sensor Location	Machine Learning Algorithms	Results
This work	Recognition of 8 grasps types with two soft-sensors	10 (2 per finger)	Contact force	Force that the prosthesis applied over the object	Fingertips	LR	20.80%
						k-NN	98.50%
						SVM	12.50%
			Angle	Angle in a joint of the finger	Finger DIP joints	DT	94.30%
						KMC	10.30%
						HC	12.50%
Konstantinova et al. [34]	A classification algorithm that distinguishes between hard and soft objects	8 (4 per finger, being 2 fingers)	Optical fiber	Force sensor Torque sensor 2 Proximity sensor	Fingertips	ZeroR	63.60%
						Perceptron	69.00%
						SVM	87.30%
Kaboli et al. [35]	To identify 20 objects of the ADLs through texture properties	5 (1 per finger)	BioTac	Contact force	Fingertips	SVM	96.00%
						PA	87.00%
						EM	80.58%
Huang et al. [36]	Recognition of gestures, object shape, size and weight	10 (2 per finger)	Optical fiber	Curvature	Along the finger	k-NN	Gestures: 97.96% Shapes: 90.81% Size: 90.79% Weight: 100.00%
						SVM	Gestures: 96.55% Shapes: 90.56% Size: 90.90% Weight: 100.00%
			Intelligent digital display pressure transmitter	Pressure	It is installed in parallel with the soft finger in each gas pressure channel.	LR	Gestures: 95.6% Shapes: 86.94% Size: 79.73% Weight: 100.00%
						KMC	Gestures: 96.83% Size: 99.37%

## Data Availability

Not applicable.

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
