# Peer review of "Soft-Sensor System for Grasp Type Recognition in Underactuated Hand Prostheses"

_sensors, 2023, doi:10.3390/s23073364_

Round 1

Reviewer 1 Report

There are actually two contributions in this paper. 

1. The usage of POF for angle and contact force measurement. This is interesting and the authors can present this part with more details. The approach and sensors used here can be also useful for soft robots.  Please check if there are some similar work in soft robots community.

2. For the grasp type recognition,  there are actually many previous works that collect data from human demonstrations and then classify them to different types. It is better to discuss the diffence here and how the new sensors used in this work can benefit the recognition.

De Souza, Ravin, et al. "Recognizing the grasp intention from human demonstration." Robotics and Autonomous Systems 74 (2015): 108-121.

El-Khoury, Sahar, Miao Li, and Aude Billard. "On the generation of a variety of grasps." Robotics and Autonomous Systems 61.12 (2013): 1335-1349.

Author Response

Dear Reviewer #1, please find attached the file of answer to reviewers 

Reviewer 2 Report

The development of an intelligent soft-sensor system to add haptic perception to underactuated hand prostheses is presented in this paper. Two sensors based on optical fiber were constructed, and the recognition experiments of eighth grasp types through the information from two low-cost POF-based force and angle sensors were tested using six machine learning algorithms. The structure of this paper is well organized. However, there are some comments needed to be addressed.

1.     Whether repeated bending of optical fiber in Figure 2 will cause errors in experimental results?

2.     It is suggested to grasp the object with the palm facing the ground. Otherwise it cannot prove whether the object can be successfully grasped.

3.     As for the introduction section on the prosthesis, it is suggested to add the more advanced research in recent years. For example, “Development of an untethered adaptive thumb exoskeleton for delicate rehabilitation assistance,” IEEE Transactions on Robotics,  2022.

4.     The grammar needs further improvement. For example, the sentence: “To validate the algorithms was used the k-fold test with a k=10”

Author Response

Dear Reviewer #2, please find attached the file of answer to reviewers 

Reviewer 3 Report

Please find attached my review

Author Response

Dear Reviewer #3, please find attached the file of answer to reviewers 

Reviewer 4 Report

L. V. De Arco et al. presented the development of an intelligent soft-sensor system to add haptic perception to hand prostheses. Two POF sensors were tested: one for finger joint angles and the other for fingertips’ contact force. Moreover, six machine learning algorithms were tested to monitor the prosthesis activity and to recognize grasp types.

This work is interesting and of merit. Here are suggestions before it can be accepted for publication.

1.     The figure captions need to revised: Figure 2 (2d is missing), Figure 3 (caption need to be corrected) and Figure 7 and 8 (both captions should have same style).

2.     The authors can briefly explain the sensing mechanism of the POF, particularly considering that the contact force and bending angle are two different physical parameters and should have different measurement approaches.

3.     The setups can be complicated with many components, making the readers difficult to read, particularly Figure 2, 4 and 5. The authors have an excellent plan to characterize the performance of the sensor individually and integrated in Figure 2, 4 and 5. These figures can be further improved and clarified for easy-reading. For instance, are the POF sensors for Figure 2 and 4 the same or different? The schematic in both should include the POF sensors, which the readers can correlate the schematics with the real pictures. Also, Figure 5 is confusing, perhaps because both the hand and object have similar colors with a bunch of optical fibers. The authors are suggested to make both figures easier to read for the readers.

4.     About Figure 6, what 0,90,180 and 270 in the rightmost caption box means? Please define them.

5.     As some of the POF sensor works present the measurement data with the power change under force or angle, is there any particular reasons the authors use the voltages instead?

6.     There are very encouraging to have six ML for the gesture recognition. Can the authors explain how certain ML has better performance than others, instead of solely presenting the results?

7.     Since one of the important parameters for a sensor is its sensitivity, can the authors address how the sensitivity can impact the outcome of this study?

8.  Some paragraphs in Introduction and Discussion are too long, and can be trimmed down.

Author Response

Dear Reviewer #4, please find attached the file of answer to reviewers 
